# Interconnectedness between Supply Chain Resilience, Industry 4.0, and Investment

**Adnan Al-Banna** [1,2] , **Zaid Ashraf Rana** [3] , **Mohammed Yaqot** [1] and **Brenno Menezes** [1,*]

1   Division of Engineering Management and Decision Sciences, College of Science and Engineering, Hamad Bin Khalifa University, Qatar Foundation, Doha P.O. Box 34110, Qatar; adalbanna@hbku.edu.qa (A.A.-B.); myaqot@hbku.edu.qa (M.Y.)
2   Department of Logistics and Supply Chain, Milaha, Doha P.O. Box 153, Qatar
3   Sindh Engro Coal Mining Company, Karachi 75600, Pakistan; zaid.ashraf@secmc.com.pk
*   Correspondence: bmenezes@hbku.edu.qa

**Abstract:** *Background:* As industry and society move towards the second decade of the deluge of advanced technologies of the Industry 4.0 (I4.0) age, introduced circa 2012, it is evident that the global economy continues to grapple with a plethora of ever-intensifying disruptions and vulnerabilities that inflict unexpected and uncontrolled damages on multiple organizational processes. These circumstances demand significant paradigm shifts, placing supply chain resilience (SCR) in the foreground of boardrooms and agendas of executive meetings. *Method:* This paper presents a bibliometric analysis of selected articles that examine the intricate interplay of I4.0 and SCR under investment constraints. Employing a funnel approach, this study delves deeper into about a hundred papers that were initially selected from a pool of approximately four thousand publications on SCR. The study thoroughly analyzes the interconnection between SCR, I4.0, and investment (INV) while classifying these articles in a structured manner, based on industry type and focus. *Results:* The primary aim of the paper is to identify trends, gaps, and potential opportunities for future research on the SCR-I4.0-INV interplays. *Conclusions:* The findings reveal that industries are converging towards the implementation of digital technologies as a strategic move to tackle unexpected, unplanned, and undesired situations. This research illuminates the needs for organizations to prioritize supply chain resilience in the face of disruptions and vulnerabilities while highlighting the potential of digital technologies to enhance their resilience, therefore ensuring sustainable growth.

**Keywords:** digital supply chain resilience; Industry 4.0; investment; bibliometric





## 1. Introduction

Over the past decade, in the wake of the so-called Industry 4.0 (I4.0) age, from economic, financial, and marketing perspectives, businesses have progressed from local barter trading entities to global cyber-market conglomerates. Hence, organizations realized there were increasing opportunities to evolve towards an efficient and resilient state as well as an abundance of threats that come together with the use of the augmented information age technologies of the I4.0 mandate. Such high-performance trading and its inherent consequences were in consonance with the evolution of Industry 3.0 (I3.0)—the digital industry of the programmed logic actions and controls—to the I4.0 revolution of the digitalization of everything, cloud and edge computing, pervasive sensing, artificial intelligence (AI), Internet of Things (IoT), and big data analysis (BDA), to name a few.

In today's industrial and societal age of the I4.0 revolution and within an unprecedented pace of an accelerated digital transformation (DT) adoption and adaptation widespread in organizational strategies, supply chain resilience (SCR) has rapidly become of interest in the form of adopting a wide range of I4.0 technology enablers to reach the desired SCR state. This intuitively implies a need of investment, whereby a proper investment balance in the resilience fitness space must be accounted for to reduce vulnerabilities without

eroding profit [1]. In the same way, organizations and businesses witness exponential and continuous shifts in their self-examination when analyzing the dynamics of the strengths, weaknesses, opportunities, and threats (SWOT) for their growth on opportunities when embracing I4.0 technologies, which comes together with awareness of inherent threats (such as cybersecurity concerns). Therefore, such an unprecedented speed of market evolution counting on investments in digitalization must be accompanied by increased resilience to combat the associated risks and uncertainties.

In terms of business resilience, lessons learned over the past decades from numerous disruptions strengthened companies' abilities to cope with such unexpected, unplanned, undesired, Un-of-Everything (UoE) situations. They are, for example, economic crises, localized epidemics such as MERS/SARS, natural disasters (e.g., Iceland's ash cloud in 2011, earthquakes, tsunamis), cyber-attacks, geopolitical conflicts (wars, blockades, etc.), and the unprecedented pandemic (the first within a digital world) of COVID-19. Consequently, with the support of cyber-physical systems and security (CPSS) solutions, agile organizations have merged (or are willing to merge) their cognizance of economy and finance concerns with efficiency and sustainability directions in line with I4.0 technologies and SCR principles for combatting risks and uncertainties they are continuously bombarded with, thus progressively enhancing businesses' performance.

Based on a bibliometric literature review framework, this paper introduces how I4.0 elements such as AI and IoT can augment SCR towards the digital supply chain resilience (DSCR). It aims to address the interconnectedness between SCR, I4.0 and investment (INV) as seen in Figure 1. We suggest that to meet targeted SCR, investment in I4.0 elements must be determined since they are constrained by the trade-off between decreasing vulnerabilities and eroding profits.

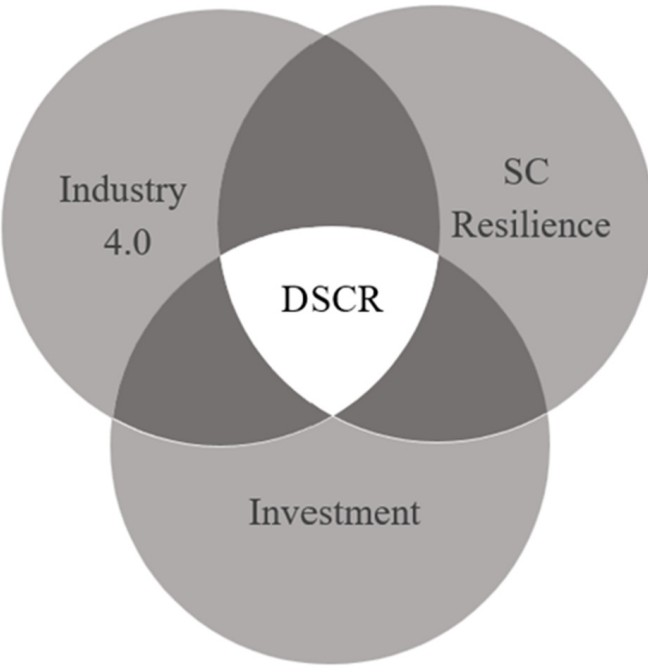

**Figure 1.** Intersection between SCR, I4.0, and investment.

The I4.0 domain in Figure 1 describes the integration of digital technologies into manufacturing, supply chain (SC) and a wide range of organizational operations. This includes technologies such as the Internet of Things (IoT), cloud computing (CC), artificial intelligence (AI), among others. I4.0 is transforming the way that businesses operate and interact with their SCs and is becoming increasingly important as companies seek to improve their DSCR and make more informed INV decisions.

The second, the SCR domain, refers to a company's ability to quickly adapt and respond to disruptions in its SC, such as natural disasters, political instability, or economic

downturns. In today's globalized economy, SCR is crucial for maintaining business continuity and reducing the risk of costly disruptions. By interconnecting I4.0 technologies with SC operations, companies can improve their SCR by gaining greater visibility into their SCs and by implementing more agile and flexible SC processes.

The third domain, the investment, is another critical factor that is closely linked to I4.0 and SCR. Investment decisions play a vital role in determining a company's ability to respond to SC disruptions and to maintain business continuity. Companies that invest in I4.0 technologies are better positioned to adapt to changing market conditions and to respond to disruptions in their SCs. By investing in DSCR, companies can reduce the risk of disruptions and improve their ability to respond quickly to unexpected events. Nevertheless, beyond an optimal resilience fitness space (RFS) region of the needed investments in I4.0, increasing SCR starts to erode company's profit, therefore reducing its sustainability of continuing as a player in the market.

The intersection of the I4.0, SCR, and INV domains is represented in Figure 1 (I4.0, SCR, and INV) as DSCR has significant implications for the future of SC management. By leveraging I4.0 technologies, companies can improve their SC visibility and gain greater insights into their SC operations. This, in turn, enables them to make more comprehensive INV decisions in order to implement more agile and flexible SC processes. By precisely investing in DSCR, companies can reduce the risk of disruptions and improve their ability to respond quickly to unexpected events.

The intersected area between the abovementioned three domains represents an organizational DSCR, which harnesses the power of I4.0 to augment an organization's ability to digitally detect, avoid, manage, and recover from disruptions and challenges. This is supposedly based on a set of conjugated investments in I4.0 elements that, when combined, meet a synergetic state in the organizational sustainability aiming a resilience state from grassroots projects in new design frameworks to shop-floor operations. By properly interconnecting I4.0 technologies with SC design and operations, companies can improve their SCR and reduce the risk of disruptions. This, in turn, can help to ensure business continuity and reduce the costs associated with SC disruptions. Furthermore, INV decisions can be made more accurately and quickly with the support of I4.0 technologies, leading to a more efficient and effective SC.

In summary, the intersection of I4.0, SCR, and INV is a critical area that businesses must focus on in order to improve their competitiveness and ensure long-term success. By investing in DSCR and leveraging I4.0 technologies, companies can improve their SC visibility, flexibility, and agility, enabling them to respond quickly and effectively to disruptions in their SCs.

This paper addresses the literature review on the interplay of SCR, I4.0 and INV, illustrating the literature review methodology (keywords and filters) in Section 2. Discussions on the SCR within (a) the COVID-19 scenario and (b) major industries, applications, and building-blocks (I4.0 elements and INVs) of the SC are presented in Section 3. The interplay between INV and DSCR is discussed in Section 4. The detailed bibliometric analysis of DSCR publications is illustrated in Section 5. Specially, two cases in I4.0 in DSCR are presented. They are the interplay of DSCR and additive manufacturing enabler, which is discussed in Section 6, and the importance of cybersecurity in the digital supply chain domain in Section 7. Finally, Section 8 presents the conclusions of the paper.

This paper contributes to the knowledge base in DSCR by (a) conducting a detailed bibliometric analysis of selected articles that cover the interplay of I4.0 and SCR under INV constraints, (b) implementing a funnel approach to analyze relevant papers addressing the interconnectedness between SCR, I4.0, and INV from an industrial perspective, and (c) classifying and analyzing these articles in a structured manner to identify trends, gaps, and potential opportunities for future research on the SCR-I4.0-INV interplays.

Overall, this paper contributes to the growing body of literature on the interplay of I4.0 and SCR and provides insights into how organizations can use digital technologies to

enhance their resilience and prepare for unexpected, unplanned, undesired, Un-of-Everything (UoE) disruptions.

## 2. Literature Review Methodology

This review paper analyzes SCR research published during the past two decades, with a focus on the interplay and interconnectedness between SCR, I4.0 and INV. It is evident, as seen in Figure 2, that SC efficiency (SCE) has attracted the most research interest over the past two decades, followed by SCR. However, research in the domains of digital SC efficiency (DSCE) and digital supply resilience (DSCR) remains insufficient.

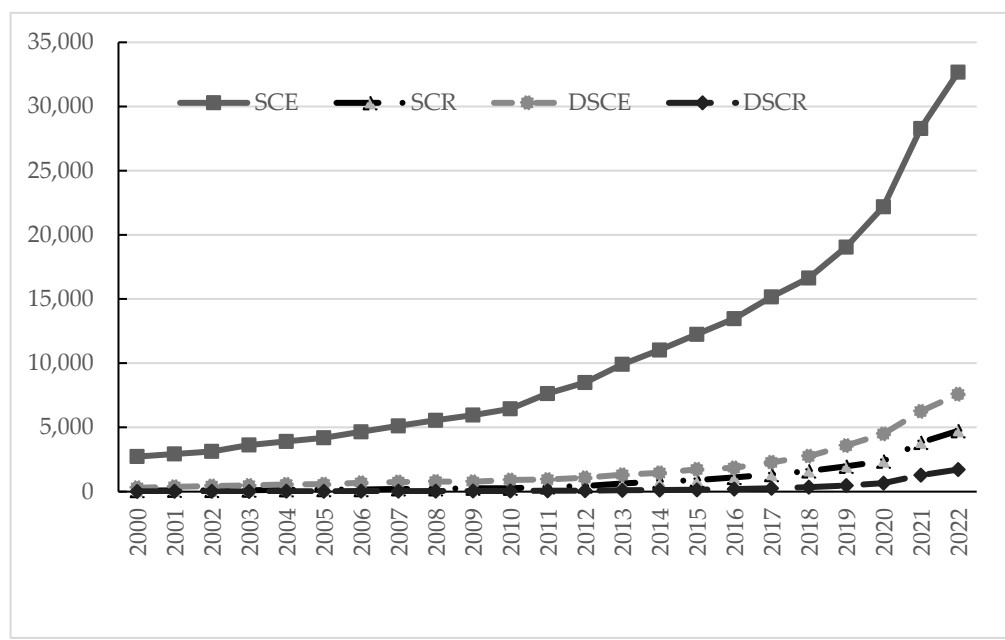

**Figure 2.** Research literature evolution in SCE, SCR, DSCE and DSCR [1].

The phenomenon highlighted in Figure 2 emphasizes the limitation of the current literature review, which is not proven to be true for academia only, as the same can be argued for the industrial domain. The current literature review only considers the conventional SC topics but remains unable to demystify topics such as DSCR, nor possible methodologies to simulate, calculate, or model the path towards I4.0 in SCR that yields the targeted DSCR. A review of all Scopus publications over the two decades between the years 2002 and 2022 revealed that about 541,634 research were published on SC, while only 10,858 research were published about DSCR, making up 2% of all the publications on SC of the same period. Furthermore, taking a deeper dive in the literature review, it was found that only 8812 studies were published about DSCR simulation and/or calculation or modeling during the same period of two decades, which is equivalent to only 1.6% of all the publications on SC of the same period.

Similarly, the industry has limited comprehension about the DSCR, its applications, and success strategies; hence, organizations appear to be hesitant to adopt I4.0 technology enablers and seem myopic regarding the optimum route towards achieving DSCR. Consequently, it is imperative to address this literature gap and provide academic and industrial professionals with the necessary knowledge, tools, and roadmaps [1] to ensure the harnessing of the power of I4.0 technologies and the achievement of efficient and effective DSCR. In this context, Zhang et al. [2] provide a comprehensive review of the current state of smart supply chain management in North America in the context of Industry 4.0. This research discusses the potential benefits of implementing smart supply chain management practices, such as improved efficiency, cost reduction, and enhanced customer service. The research also identifies some of the key challenges and barriers to adoption, including issues related

to data privacy and security, interoperability, and workforce training. However, the authors also highlight the limitations of the current literature on smart supply chain management from a research perspective. One limitation is that most of the existing research has focused on the development of individual technologies, rather than the integration of these technologies into a comprehensive supply chain management system. The authors suggest that more research is needed to develop integrated solutions that can effectively leverage the benefits of multiple technologies.

In addition, Zhang et al. [2] highlight another limitation, where most of the existing research has focused on the use of smart supply chain management in specific industries, such as manufacturing and logistics, rather than across industries. The authors suggest that more research is needed to explore the potential benefits of smart supply chain management in a wider range of industries. The authors also note that there is a need for more research on the impact of smart supply chain management on firm performance and competitiveness. While there is some evidence that smart supply chain management can lead to improved performance, there is still a lack of understanding of the specific mechanisms through which this occurs.

In the same connection, Alkaabneh et al. [3] highlight the lack of a comprehensive approach in the existing literature that considers both equity and efficiency measures in non-profit settings. The authors note that previous research has largely focused on either efficiency or equity objectives and has not sufficiently considered the interdependence of these objectives in non-profit settings. In particular, the research highlights the limited attention given to the issue of equity in non-profit settings, despite its importance in ensuring fair and just distribution of resources. The authors argue that the existing literature has mainly focused on efficiency measures such as minimizing transportation costs, without taking into account the distributional implications of these measures. Another limitation of the existing literature identified by the research is the lack of consideration of demand uncertainty in routing and resource allocation in non-profit settings. The authors note that demand uncertainty is a critical factor that affects the ability of non-profit organizations to achieve their objectives; yet, it has received limited attention in the literature.

In response to these limitations, Alkaabneh et al. [3] propose a comprehensive approach that considers multiple objectives, including equity, efficiency, and demand uncertainty, in routing and resource allocation in non-profit settings. However, the authors acknowledge that their proposed approach is not without limitations, such as the assumption of perfect information and the simplifications made in the mathematical model.

Furthermore, Nakandala et al. [4] propose a serial mediation model that links I4.0 technology capabilities to incremental innovation through resilience as a mediator. The research finds that I4.0 technology capabilities positively influence resilience, which in turn leads to incremental innovation in Australian manufacturing firms. The research also identifies the specific I4.0 technologies that are most significant in driving incremental innovation, including BDA, IoT, and CPS. A key contribution of this research lies in its emphasis on literature review limitations. The research notes that previous research on I4.0 technology adoption has largely focused on the factors that influence the adoption decision and the benefits that can be derived from the adoption of new technologies. However, there is a limited understanding of the mechanisms through which I4.0 technology capabilities affect incremental innovation, particularly in the context of non-Western developed countries such as Australia. Additionally, the research's contribution to the literature and its findings suggests that I4.0 technology capabilities, resilience, and incremental innovation are interconnected and should be considered holistically in the context of manufacturing firms.

In a similar context, Alkaabneh and Diabat [5] highlight a significant limitation in the existing research, which is the lack of multi-objective models in the supply chain field. Most previous studies pertaining to the healthcare industry have focused on single objectives, such as cost efficiency, and have not considered other factors such as service quality and patient satisfaction. This limitation can be attributed to the complexity of the home healthcare delivery system, which involves multiple stakeholders with varying

objectives and preferences. To address this limitation, the authors propose a multi-objective model that considers three objectives: cost, service quality, and patient satisfaction. They suggest that this model can help decision-makers to achieve a balance between these objectives and make informed decisions about home healthcare delivery. Additionally, the authors propose two solution algorithms, a branch-and-price algorithm and a two-stage meta-heuristic algorithm, to solve the multi-objective model. The proposed model and algorithms contribute to the literature by addressing the limitations of previous studies and providing decision-makers with a comprehensive framework for home healthcare delivery. Nevertheless, the authors acknowledge that their proposed model and algorithms have limitations, such as the assumption of fixed patient demands and the simplification of some real-world factors. Therefore, they suggest that future research should explore other multi-objective approaches and compare their performance with the proposed model to provide more comprehensive and robust solutions for home healthcare delivery.

Next, Fernando, et al. [6] investigate the impact of information system security practices on cyber supply chain risk management and performance of Malaysian manufacturing companies in the I4.0 era. The research employs a quantitative approach by surveying 202 manufacturing firms and analyzing the collected data using structural equation modeling. The authors highlight several limitations in the literature review, such as the lack of studies that have investigated the relationship between information system security practices and cyber supply chain risk management in the context of I4.0, especially in Malaysia. The authors also point out the scarcity of studies that have tested the role of information system security practices in mitigating the impact of cyber supply chain risks on organizational performance. The research's findings reveal that information system security practices positively affect cyber supply chain risk management, which in turn positively impacts organizational performance. The research also emphasizes that the implementation of information system security practices significantly moderates the relationship between cyber supply chain risk management and organizational performance.

In the same connection, Alkaabneh, et al. [7] investigate carbon reduction strategies of fruit supply chains using a systems approach. The research identifies the main sources of carbon emissions and proposes potential policy options that could be implemented to reduce emissions, including carbon taxes, technology innovation, and land sparing. The research uses a case study of the Australian apple and pear industry to demonstrate the effectiveness of the proposed policy options. In terms of the literature review limitations, the research highlights that the existing literature on carbon reduction strategies in the fruit supply chain has largely focused on individual stages of the supply chain, such as production or transportation. However, the authors argue that a systems approach is required to address the complexity of the fruit supply chain and to identify the most effective carbon reduction strategies. Additionally, the authors note that most of the existing literature has focused on developed countries, and there is limited research on carbon reduction strategies in developing countries. Therefore, the study makes a valuable contribution by proposing a systems approach to carbon policy and by providing insights into potential policy options for reducing carbon emissions in the fruit supply chain.

Considering this, in the next section, the paper provides an overview of supply chain resilience perspectives in a number of critical industries followed by a robust and structured bibliometric analysis.

The review methodology of this paper examines a number of previous structured review research studies. Firstly, research that provides a systematic literature review on the impact of AI on workplace outcomes is presented [8]. This research offers a comprehensive review of existing research with the objective of identifying and analyzing relevant literature on the topic. The authors categorized the outcomes of AI on workplace processes into three levels: individual, team, and organizational. This work identifies positive and negative impacts on each level and states that the impact of AI on workplace outcomes depends on several factors, including the level of employee involvement in AI implementation, the degree of task complexity, and the level of trust and transparency in

the AI decision-making processes. The authors also propose a multi-process perspective to understand the impact of AI on workplace outcomes, which includes cognitive, affective, behavioral, and social processes. This perspective emphasizes the importance of examining the impact of AI on various processes within an organization to fully understand its impact on workplace outcomes.

In the second structured review research pointed out in this paper, the work provides a systematic review on R&D internationalization and innovation [9]. The research reviews several electronic databases and 129 peer-reviewed articles are included in the final sample. The research proposes an integrative framework for understanding the complex relationship between R&D internationalization and innovation, which includes five dimensions: the degree of internationalization, the type of innovation, the organizational context, the geographical context, and the industry context. The framework emphasizes the need to consider these dimensions when studying the relationship between R&D internationalization and innovation. The authors also identified several future research directions, including the need to examine the role of specific organizational factors in the relationship between R&D internationalization and innovation, the need for longitudinal studies, and the need to consider the impact of different types of R&D internationalization on an organization's innovation.

In summary, while both research studies in [8,9] employ a systematic review methodology, they differ in their research topics and outcomes. The former research [8] provides insights into the impact of AI on workplace outcomes, while the latter study [9] contributes to the understanding of the relationship between R&D internationalization and innovation.

The abovementioned approaches were considered a leverage in fine-tuning this research paper methodology. This bibliometric review paper analyzes relevant research that is published during the last twenty years during the period between 2002 and 2022, with a focus on the niche area of the intersection between SCR, I4.0 and INV. The literature review is conducted by performing keyword searches in Scopus and Web of Science (WOS) databases. The methodology adopted a multi-level keyword search based on a funnel approach.

In level 1, the search query "supply chain" AND "resili*", where "AND" returns results with both the keywords in the title, abstract or keywords; this level yielded 3699 papers.

In level 2, the search query "supply chain" AND "resili*", AND "Industry 4.0" OR "digita*" OR "digital transformation" is used, which yielded 389 papers. In this and subsequent levels, asterisks (*) are used with the objective of sourcing multiple words that contain the root text, for example "resili*" captures words such as resilience, resilient, and resilience. In the same connection, generic and full keyword searches such as "Supply Chain", "Resilience", and "Investment" separated are avoided as they could yield large numbers of irrelevant results.

In level 2*, conference papers are filtered out, and only journal papers published in the English language are considered; this created a shortlist of 201 papers.

In level 3, the search query of level 2 is used in addition to "investment" OR "econom*" OR "financ*"; this yield 141 papers.

In level 3*, the level 3 results are limited to only articles in journals and in English, which yield 67 papers. However, 8 more articles that do not appear in the keyword search are added during the review process, making the final list a total of 75 papers.

The review revealed that articles are broadly distributed under different categories as per their area of concern. In this paper, a comprehensive bibliometric analysis is conducted; in addition, the COVID-19 impact on SCR is addressed, as well as a detailed review of SCR with respect to hand-picked industries. The industries are chosen based on their perceived importance to global economies, namely maritime, food, automotive, and logistics. The review utilized research that is sourced from Scopus and Web of Science, as illustrated in Figure 3. The literature review indicated that the Scopus database appeared to be more comprehensive than that on Web of Science with regard to this research subject, although

duplicates were identified and the same was considered while conducting the literature review funnel.

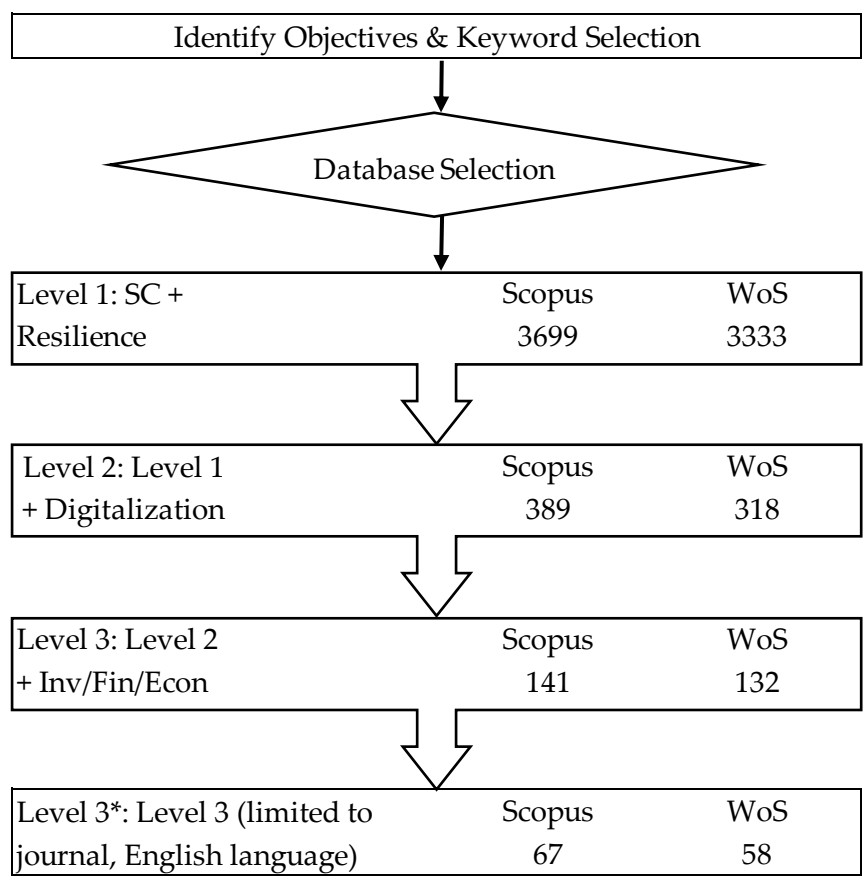

*: Limiting search to English language and journals

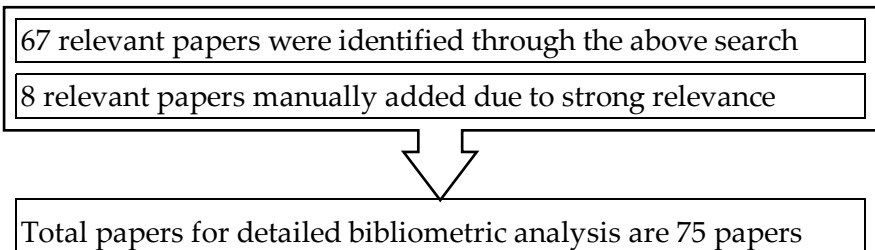

**Figure 3.** Graphical representation of Scopus and WoS research sources comparison.

## 3. Supply Chain Resilience: COVID-19 Scenario, Major Sectors, and Main Building-Blocks

### 3.1. Building Supply Chain Resilience in the COVID-19 Scenario

The global crisis following the COVID-19 pandemic is regarded as the largest disruptive event in recent world history [10]. Accordingly, the academic community scrambled to take crucial steps to research the mitigation of its impacts on SCs. Outbreaks such as COVID-19 are positioned as a risk to the SC as they lead to long-term disruptions, uncertainty, and disruption propagation. The authors of [10] performed a simulation-based study and presented an approach to aid decision-makers in SC planning to mitigate risks from outbreaks. In the early stages of the COVID-19 pandemic, various research aims to examine the impacts of SC disruptions due to the COVID-19 pandemic [10–12] and it is predicted that companies working on end-to-end DT and having operational resilience will be successful in the post-COVID-19 scenario. Large-scale usage of the internet and communication technologies along with financial contingency plans are vital for the resilience

of an organization during different stages of a crisis such as COVID-19 [13]. Long-term business sustainability can be achieved through the implementation of I4.0 practices in the face of COVID-19-like problems.

Success and resilience resulting from the adoption of digital technologies are not just limited to large organizations and large SCs. Smallholder farmers in developing countries can increase their productivity and resilience by using digital technologies [14]. Agricultural SCs were exposed to the COVID-19 disruptive risks such as those related to SC elements of supply, demand, logistics, finance, operations and management, policy, and environment. On the other hand, small and medium enterprises (SMEs) could increase their resilience to cope with future uncertainties during the COVID-19 pandemic by transforming their businesses in line with I4.0 practices [15]. The effect of COVID-19 on South African omnichannel SCs is empirically studied [16]. The author identifies suppliers' inability to meet demand surges resulting from the migration of customers to online channels and front- and back-end operational constraints as significant disruptions. However, fluent strategic decision-making, long-term planning, and agility to unlock scale, and investment (INV) capacity serve as effective countermeasures for omnichannel retailers against these disruptions.

Another research focused on six Southern Italian SMEs who turned the COVID-19 crisis into an opportunity by leveraging digital technologies, flexible financial resources, collaboration with research institutes, and strategic agility [17]. Conversely, it was identified that disruptions in the SC and shortages of cash flow serve as serious limitations for SMEs; however, digitalization gained momentum, which provided success and profitability when used in a proper way. The digital transformation economy aids to enhance economic growth and reduce SC disruption risks; furthermore, it assists in the organized functioning of society during the pandemic [18,19].

Diversification of production sources and improving conventional manufacturing capacities can give assistance to bridge SC vulnerabilities [20]. Digitalization can help improve production systems during COVID-19 disruptions [21]. SCR and sustainability can be improved by taking measures that include SC regionalization, agility, supply network diversification, visibility, collaboration, transparency, practices of circular economy, and the usage of smart technologies [22].

SCR is directly proportional to capacity and inversely proportional to ripple effects of disruptions. For example, it was identified that significant traits of disruptions could assist organizations mitigation strategies in order to develop a resilient SC, find weaknesses to be identified and rectified, and foresee the capability of anticipation of disruptive events to be developed [23]. The same could be considered building blocks for SC managers to manage SC disruptions using digital technology [24]. Furthermore, studies emphasize that digitalization and automation are poised to play crucial roles in SC transformation in the post-COVID-19 world [25,26]. Moreover, to investigate the impact of I4.0 technologies on supply chain (SC) management, the SC transformation was analyzed using lean thinking principles [27]. The study utilized the SC operations reference (SCOR) model to evaluate the improvements that arose from the application of these technologies [28]. The identification of critical factors that promote the sustainability of supply chains during and post-COVID-19 was the focus of another research as found in Yin at al.'s work [29]. Through the use of the stepwise weight assessment ratio analysis (SWARA) method, the study determined that the supply chain network viability (CSV) criterion is key to enhancing the relationship between buyers and suppliers, which in turn contributes to the survivability of sustainable supply chains (SSC). The resilience of the SCs of vaccines and other pharmaceuticals is of critical importance during an outbreak. The SCR of vaccines during COVID-19 is studied in [30]. In this context, the optimal system performance can be augmented by utilizing AI, digital twins, and stress tests. Similarly, it is perceived that healthcare industry ecosystem can enhance its SC ecosystem through the adoption of I4.0 technologies and making the procurement of pharmaceuticals more resilient during the pandemic era [31,32].

### 3.2. Supply Chain Resilience and the Maritime Shipping Industry

The maritime industry has taken measures over time to develop SCR. Increased digitalization, collaboration, better resource allocation, building port network hubs, capacity sharing, and cross-port INVs are some of the focus points to improve maritime SCR. Academia acknowledge the scarcity of literature available on SCR of the maritime shipping industry even though this transportation modal contributes to a significant share of the economy. The authors highlight the need for quantitative research in the maritime SC and the requirement for analytical frameworks towards shipping resilience in the wake of climate change [33]. Several authors have discussed case studies of shipping companies located in various geographic locations to identify the challenges posed by disruptive events and the role of DT in this context [34]. A simulation-based framework has been developed to test the proposed approaches and to present the framework as a tool for SCR. To expand the scope of the risk management approaches, the authors highlighted the need to couple their developed framework with a cost-evaluation model, whereby a network model for the ports of Shanghai and Los Angeles/Long Beach was developed in [35]. The research uses the model to evaluate resilience metrics of tonnage, time, and cost resilience. The authors studied the impact of resource allocation, diversity, and collaboration to enhance resilience levels and suggest several schemes to improve resilience and adaptive capacity to overcome natural and man-made disruptions in marine transportation systems (MTS).

In addition, statistical data from Chinese ports from 2005 to 2017 were analyzed with respect to the container port shipping network resilience (CPSN) [36,37]. A framework was used to analyze the performance of several network indicators before and after a deliberation or random attack. The research report found that the resilience of CPSN can be increased by investing in/adding port hubs. The following are some of the key points of the research: (a) the CPSN is a complex system that is vulnerable to disruptions; (b) a deliberate or random attack can significantly reduce the resilience of the CPSN; and (c) investing in/adding port hubs can increase the resilience of the CPSN.

The research findings have important implications for policymakers and stakeholders in the shipping industry. The findings suggest that it is important to invest in the resilience of the CPSN in order to mitigate the risks of disruptions.

Similarly, academia proposes strategies such as capacity sharing, and protective cross-port investment through coalition formation to improve the resilience of individual ports. The financial crisis of 2008 to 2009 and COVID-19 have emphasized the importance of enhancing resilience of container shipping and ports. The academic knowledge base also recognizes joint capacity management through alliances as a robust strategy for the shipping industry during the pandemic [38,39].

Focusing on container shipping, the author reports the consequences of the outbreak on the global SC during a period of 1.25 years (2020 to the 1st quarter of 2021). The paper stresses the need for the digitalization of maritime shipping SCs as a measure of resilience against disruptions by discounting human error from the functioning of SCs. TradeLens is introduced in the paper as a pioneering digital blockchain-based platform, developed by Maersk and IBM, for secure digital sharing of shipping data and documents to simplify and speed up trade workflows. Furthermore, the author predicts that the future induction of 5G to 6G communication technology in the maritime industry will improve resilience by speeding up communication. Similarly, there is an increasing scientific interest in the I4.0 technologies for SCR. They present a concise guiding matrix for the implementation of I4.0 technologies to address SC vulnerabilities (SCV). The authors report four research gaps that include the need to carry out more research to study how I4.0 could contribute to developing SCR; the need to develop approaches to measure SCR capabilities and vulnerabilities interactions, provide descriptions of SCR terms, and the potential of work on companies' security, financial strength, and financial position; and some academic knowledge base proposing strategies such as capacity sharing, and protective cross-port INVs through coalition formation to improve the resilience of individual ports [40,41].

### 3.3. Supply Chain Resilience and the Food Industry

Resilience and smooth functioning of food SCs are crucial to avert risks of hunger and famine, especially during extended and unpredictable SC disruptions. In this regard, the transformation of conventional operations of food SCs into its DT (based on investments) is mandatory for food SCR. In this connection, Ref. [42] explores the impact of the COVID-19 pandemic on meat supply chains. It discusses the challenges faced by the industry, including labor shortages and disruptions in production and distribution. The study highlights strategies employed to mitigate the impact, such as safety measures and technological advancements. It emphasizes the need for resilience and collaboration in supply chain management for future crises. In a similar connection, research focused on the meat industry while discussing the impact of COVID-19 on food SCs concluded that the pandemic sped up the adoption of digitalization and automation within the food SCs [43]. The meat processing, vegetable, and fruit production sectors suffered from major workforce disruptions during COVID-19, in addition to stresses on the critical importance of augmenting organizations' capabilities with respect to adaptability and flexibility especially in such testing times. The research made reference to the fact that online delivery systems helped food SCs survive during and after the pandemic, and has emphasized the importance of INV in automation and digitalization. In the same context, academia acknowledges that companies that employ excellent technological innovation in their production processes were the least affected during the outbreak. Moreover, a partnership between the public and private sectors to develop sustainable and resilient food SCs should be made and cognizant investment in I4.0 is critical in order to develop a self-reliant and resilient agro-food sector [44–47].

### 3.4. Supply Chain Resilience and the Automotive and Logistics Industries

Capitalizing on the sensing, calculating, and actuating (SCA) cycle encapsulated within AI and industrial IoT (IIoT) capabilities, car manufacturers are racing to introduce the first fully autonomous car. As an example, Tesla introduced the first self-driven car to the market; nonetheless, the underlying technology is far from being ready for complete self-operation.

Academia highlights reasons for the skepticism because the United States National Highway Traffic Safety Administration (NHTSA) has raised concerns about Tesla's claims regarding their cars' readiness to hit the roads without human drivers. It is noteworthy to highlight that although NHTSA has earlier awarded Tesla the highest safety score for occupant safety, NHTSA has recently investigated fifteen crashes since the year 2016 involving Tesla vehicles equipped with autopilot. In addition, European crash test officials at European New Car Assessment Programme evaluated Tesla's Model-3 with a moderate score and regarded the name "autopilot" to be inadequate because it misleads towards complete automation, although this is not the case. German courts have banned Tesla from repeating misleading claims about the car's self-driving capabilities [48–50].

In the same connection, Elon Musk was quoted stating that the self-driven car's accuracy level in 2019 was at 98%, and it needs to be at 99.999% to be extremely reliable. Although 98% is usually considered a high percentage, in the transportation industry, whether air, sea, or land, let alone space, 98% is perceived to be considerably low. Such a percentage translates that out of each 100 families driving on a Monday morning to school, two will have an accident; thus, the key hindrance to Musk's self-driven car program is the trustworthiness aspect of Tesla's autonomous eco-system. The above illustrates that the market seeks (1) autonomous systems with the complete eco-system of smart sensors, augmented by artificial intelligence, and smart actuators; (2) improvement in technology (for a deployed stage of I4); and (3) regulations on the trustworthiness and reliability of current eco-systems (to start a belief stage of the I4 deployment). In the context of car design/style SCs, the Italian and French automotive companies have performed better than their competitors during a financial crisis because of several factors including production digitalization, and highly skilled employees, along with solid financial backing and good

relationships with clients. In a systematic literature review, the research comprehensively analyzes the link between I4.0 and SCR by using a test case of the automotive industry disrupted by COVID-19. They report that cyber-physical systems, additive manufacturing, and big data analytics are among the most effective I4.0 technologies for SCR [49–51].

Furthermore, I4.0 makes it possible to take proactive measures of risk management before the occurrence of a disruptive event. There is a need to build resilient SCs in the logistics business due to future social and demographic developments and climate change. New digital technologies, collaboration, integration of SCs, and sharing economy serve as potential solutions for sustainability transitions of logistics service providers. The path to achieving sustainability can be shortened and costs can be cut by using the experiences of successful organizations and benchmarking their data. Furthermore, the impact of the pandemic COVID-19 on the airline and automobile SCs was analyzed using both qualitative and quantitative methods to access short- and long-term response strategies. To mitigate disruptive risks, the automobile industry perceives advanced usage of I4.0 technologies and develops local supply sources. Both industries perceive the role of big data analytics for the real-time analysis of information on SC activities [52–54].

The challenges of digitalization of logistics in developing countries are unique. A conceptual framework, which shows relationships among information systems, AI, and SC disruption reports that using technology-based infrastructure can help to reduce the impact of SC disruptions. In the same connection, the Indian logistics sector uses the Bayesian best–worst method to prioritize the barriers to innovation. The top five barriers include high INV costs, low financial resources, insufficient internet connectivity, deficiency of IT infrastructure, and ambiguities about financial benefits from digital INV. In the same context, research reports that the adoption of I4.0 technologies to mitigate disruptions resulting from contiguous outbreaks, workforce management is another effective strategy to manage the disruptive effects. Similarly, the logistics service providers (LSPs) creatively managed their resources to combat operational and financial due to COVID-19. In another part of the world, the impact of COVID-19 on a Turkish automotive manufacturer was examined and the research reveals that in order to survive in a volatile, uncertain, complex, and ambiguous environment, it is imperative to redesign the SC by including recovery plans that can help to make it resilient [55–59].

## 4. Investment and Digital Supply Chain Resilience

Investment and its associated financial decisions are critical concerns for all organizations, whether these are profit or non-profit organizations. To ensure the right INV decisions are made, a wide range of parameters are to be carefully analyzed, which include—but are not limited to—cash flows, cash stocks, assets, and liabilities, returns on INV, pay-back period, net present value of the invested funds, alternative avenues of potential expenditures, cost of opportunity lost, risks of status quo, market conditions and future forecasts, among others [60]. In the same connection, the decision-making process of critical capital expenditure (CAPEX) and addressed the decision of installing wind energy capacity, which has witnessed a rapid increase in the recent few years. It is argued that understanding the impact of different deployment factors on the overall cost of a particular INV is pertinent toward benchmarking the potential of different INV decision alternatives. A similar approach could be adopted to evaluate various options of potential I4.0 technology enablers by evaluating the best INV opportunities to be considered [61].

Furthermore, within the transportation infrastructure domain and its associated supporting eco-system, in order to be resilient, its relevant decision makers and stakeholders must understand the relationship between pre-event INVs and post-event recovery success [62].

In the same context, research emphasizes the importance of increased INV in SC digital tools in order to achieve higher levels of SCR; the same is attributed to the multiple advantages that I4.0 technology enablers provide which include higher visibility, anticipation, collaboration, and other crucial elements that pave the roads for increased DSCR [63].

## 5. Bibliometric Analysis of Supply Chain Resilience Publications

In this section, we perform a comprehensive bibliometric analysis, which encompasses data from citation meta-data of the aforementioned 75 articles using the biblioshiny web-interface application of the "bibliometrix" R-package [64]. Figure 4 shows the distribution of articles in the top 10 journals, regarded as the most relevant sources.

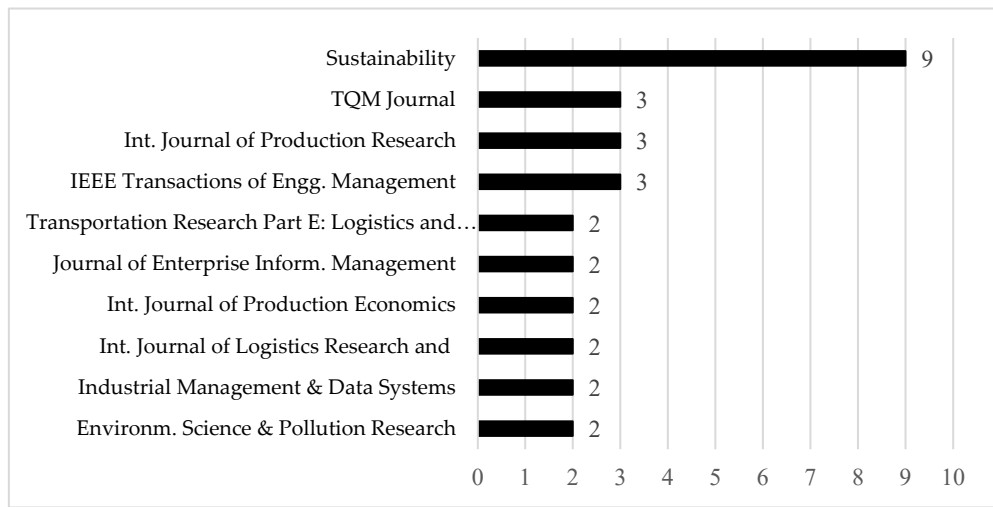

**Figure 4.** Distribution of articles in the top 10 journals, regarded as the most relevant sources.

Figure 5 shows the distribution of articles in the top 10 journals, which are regarded as the most relevant sources. The *Sustainability* journal tops the list with 9 publications, which makes up 12.3% of total publications. The journals which are in second place, each with 3 articles, include *IEEE Transactions on Engineering Management*, *International Journal of Production Research*, and *TQM Journal*.

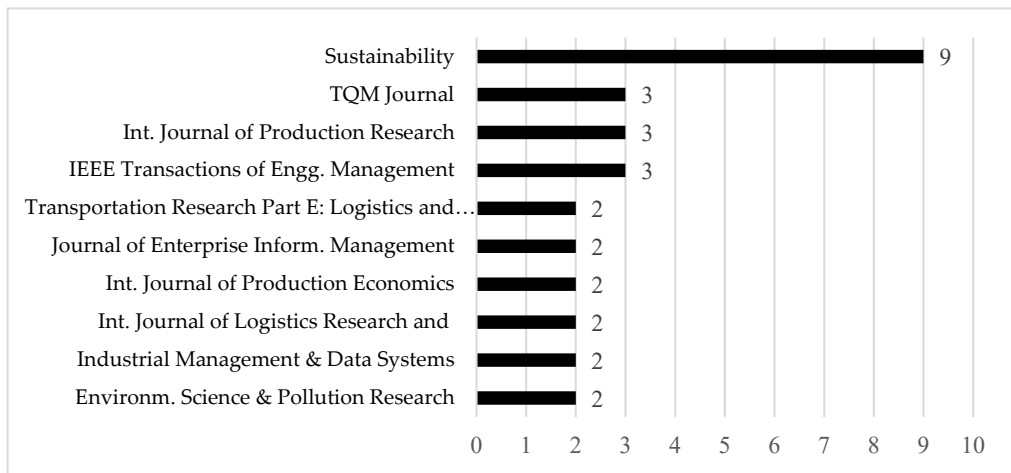

**Figure 5.** Distribution of articles in the top 10 journals.

The top 10 most cited local sources are presented in Figure 6. Local citations refer to the number of times a journal/publication/author in this paper has been cited by other journals/publications/authors of this paper. It was found that the *International Journal of Production Research* and the *International Journal of Production Economics* are the top two journals with local citations.

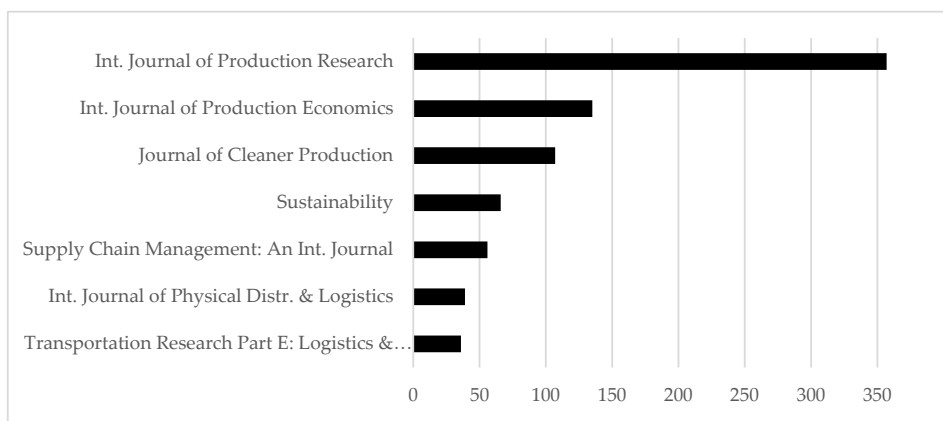

**Figure 6.** Top 10 most local-cited sources.

Figure 7 shows the top ten authors based on their h-index. The h-index measures both the quality and quantity of publications by an author by comparing publications to citations. It was found that the maximum h-index achieved by the top author is 4.

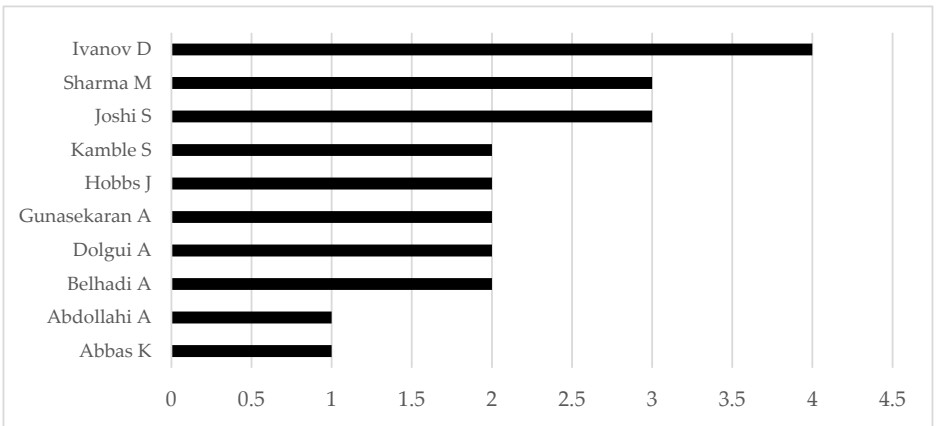

**Figure 7.** Author impact by a measure of h-index.

Figure 8 shows an intensity map of the number of scientific productions by country. Dark blue colored areas have the highest output, whereas the fading blue color reflects countries with lower scientific output.

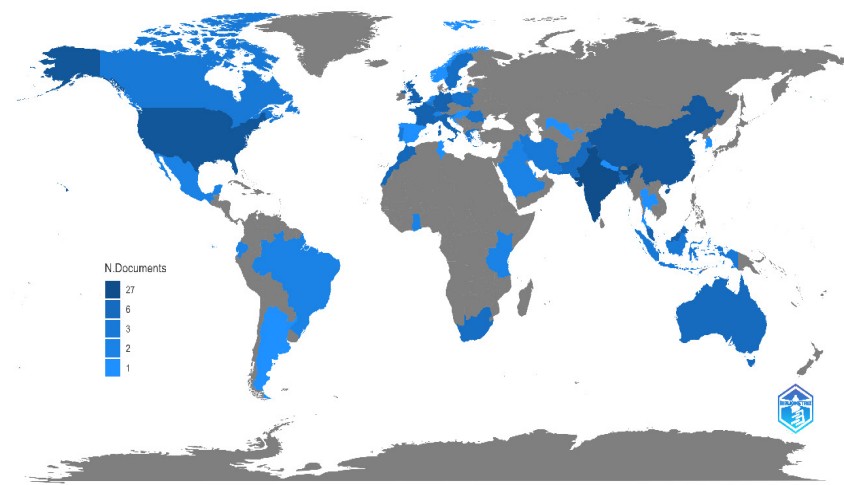

**Figure 8.** Intensity map of scientific productions by country.

Figure 9 plots the most global citations of the top 8 articles. Author, year of publication, and journal name are plotted on the *y*-axis versus the number of documents on the *x*-axis. The author Ivanov, D., has the highest number of citations for his article "Predicting the impacts of epidemic outbreaks on global SCs: A simulation-based analysis on the coronavirus outbreak (COVID-19/SARS-CoV-2) case".

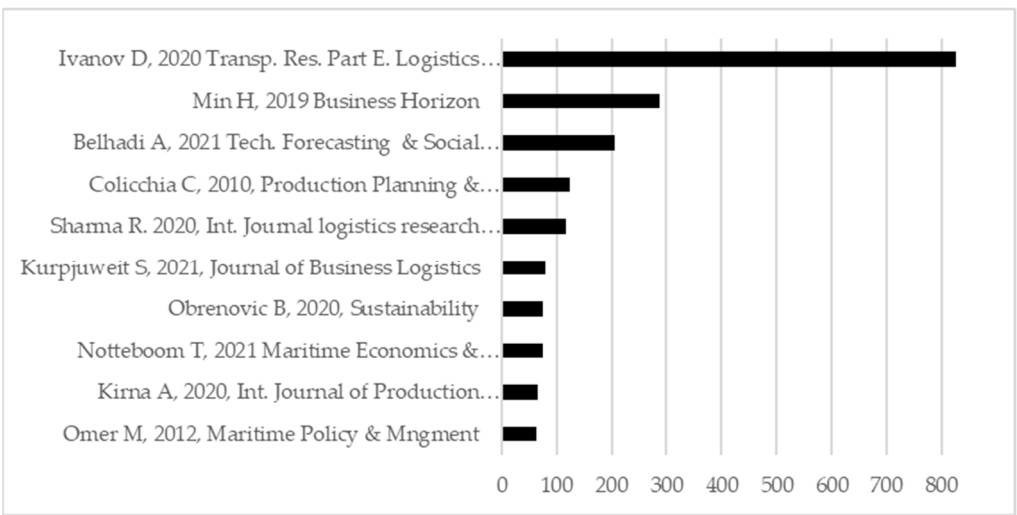

**Figure 9.** Most global citations [11,13,34,38,54,65–67].

A tree plot of the nine most frequent keywords is shown in Figure 10. For each word in the tree plot, its frequency and relative percentage are also noted.

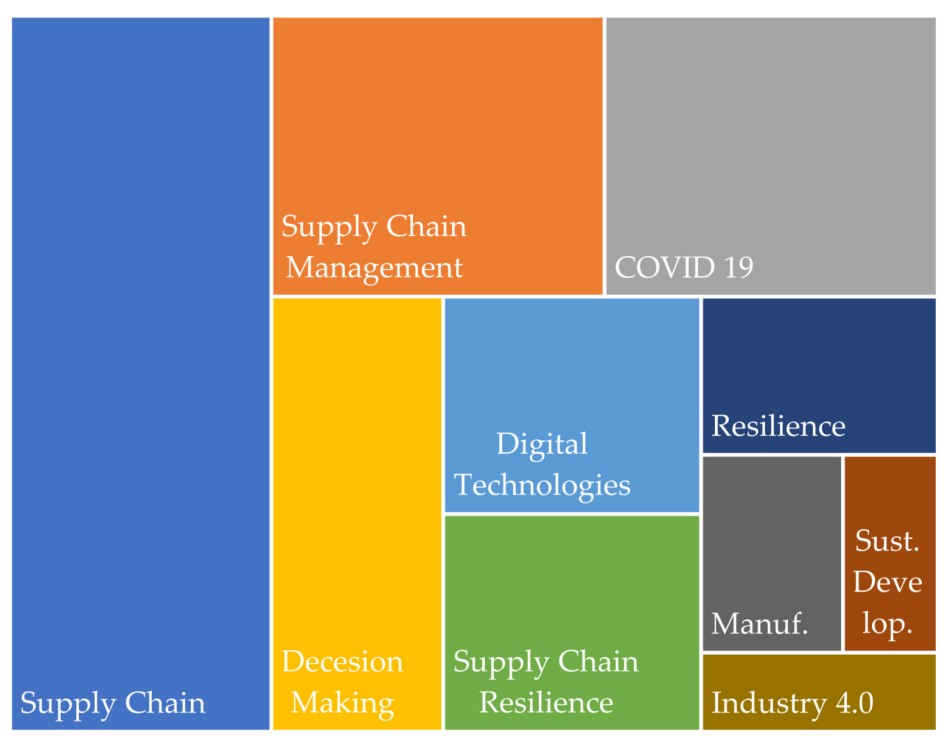

**Figure 10.** Tree plot of nine most frequent keywords.

The co-occurrence network is presented in Figure 11. Keywords from referenced journals are presented as nodes and each co-occurrence of a pair of keywords is presented as a link. The weight of a link (thickness in the figure) increases as the number of co-

occurrences of a keyword increases in multiple papers. The resulting network represents cumulative knowledge that helps to obtain an important understanding of knowledge components [68]. Six clusters (shown in distinct colors) appear in Figure 11. The orange cluster is the largest, which shows the highest number of publications with a central focus on SCs and SCR. Purple and green clusters are the second and third largest, which focus on central themes of SC management and COVID-19, respectively.

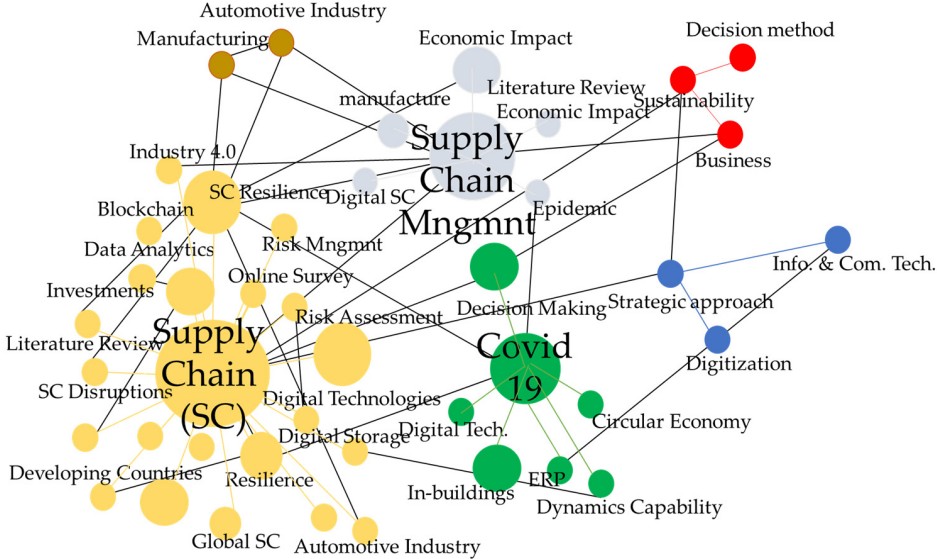

**Figure 11.** Co-occurrence network of keywords plus.

Figure 12 shows a thematic map of the relevant literature. The thematic map is segmented into four quadrants. The motor themes, situated in the upper-right quadrant, exhibit high centrality and density. These themes are considered mature and significant for the research field. On the other hand, the basic themes found in the lower-right quadrant have high centrality but low density. Although they are important to the research field, they are not as fully developed as the motor themes. However, they are general topics traversing across various research areas of the field. The lower-left quadrant contains emerging or declining themes. They have both low density and centrality, fringe themes that are weakly developed. The upper-left quadrant has very specialized/niche themes. They have high density and low centrality, which points to their limited significance for the research field [69].

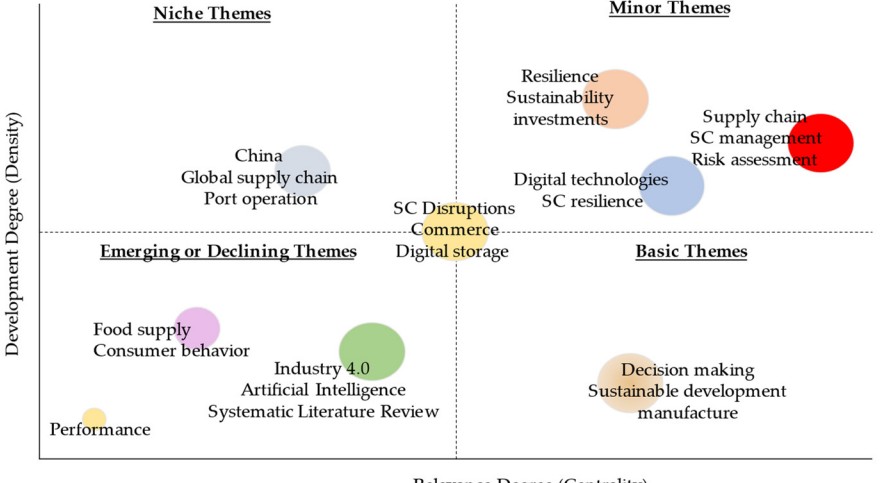

**Figure 12.** Thematic map.

It was found that research themes relating to digital technologies, COVID-19, and sustainability are motor themes with SCR, SC management, and resilience. Niche themes include global SC and port disruptions. Emerging or declining themes are I4.0, AI, food supply, and consumer behavior. Themes that have equal centrality and density are SC disruptions, commerce, and digital storage.

Further bibliometric analysis could be considered to examine the authors' co-citation and collaboration networks utilizing the Louvain clustering algorithm [70].

Table 1 summarizes the literature review papers with respect to the relevant industries based on their perceived importance to global economies, namely maritime, food, automotive, logistics, among others.

**Table 1.** Summary of the surveyed literature.

| References | Article Type | | Industry/Focus Area | Main Idea | Methodology | | | I4.0 Technologies Used/Recommended |
|---|---|---|---|---|---|---|---|---|
| | Research | Review | | | Conceptual/ Theoretical | Empirical | Mathematical/ Simulation | |
| [34] | ✓ | | Maritime | SCR in global sourcing context | | | ✓ | ✓ |
| [35] | ✓ | | Maritime | Maritime transportation resilience | | | ✓ | |
| [50] | ✓ | | Automotive | Strategies for facing a crisis | | ✓ | | ✓ |
| [52] | ✓ | | Logistics | Logistics business transformation for sustainability | ✓ | ✓ | | ✓ |
| [65] | ✓ | | Business | Enhancing SCR | ✓ | | | ✓ (Blockchain) |
| [71] | ✓ | | Manufacturing | SC agility and resilience | | ✓ | | ✓ |
| [72] | | ✓ | Manufacturing | Lean, Agile, Resilient, and Green (LARG) Manufacturing | ✓ | | | ✓ |
| [33] | | ✓ | Maritime | Impact of port disruption on maritime SC | ✓ | | | |
| [73] | | ✓ | Business | Competitive advantages through digital innovation | ✓ | | | ✓ |
| [7] | ✓ | | Business | COVID-19 outcomes for global SCs | ✓ | | | ✓ |
| [12] | ✓ | | Business | Predicting the impacts of COVID-19 on global SCs | | | ✓ | ✓ (Digital twins) |
| [74] | ✓ | | Business | Ripple effect quantification | | | ✓ | |
| [75] | ✓ | | Business | Organizational adaptation in growing risks | ✓ | | | |
| [14] | ✓ | | Business | Operational sustainability and productivity during COVID-19 | ✓ | | | ✓ (ICT, Intranet) |
| [16] | ✓ | | Agriculture | Resilient post-COVID-19 SCs | ✓ | | | ✓ |
| [17] | ✓ | | Furniture | SMEs surviving the COVID-19 | | ✓ | | ✓ |
| [76] | ✓ | | Business | SC quality management systems | | | ✓ | ✓ |
| [66] | ✓ | | Agriculture | SC risks | | | ✓ | ✓ |
| [54] | ✓ | | Automotive | SCR to the COVID-19 outbreak | | ✓ | ✓ | ✓ (Big Data) |
| [77] | ✓ | | Additive Manufacturing | SCR for managing the ripple effect in I4.0 | | | ✓ | ✓ |
| [38] | ✓ | | Maritime/Ports | Disruptions and resilience in global container shipping and ports | ✓ | | | |
| [30] | | ✓ | Healthcare | SCR—modeling approaches | ✓ | | | ✓ (AI, digital twins) |
| [56] | ✓ | | Automotive | Coping with SC disruptions | | ✓ | | ✓ (AI) |
| [58] | ✓ | | Logistics | SCR during COVID-19 | | ✓ | | ✓ |
| [42] | ✓ | | Food | Meat SCs during the pandemic | ✓ | | | ✓ |
| [43] | ✓ | | Food | SCR during the pandemic | ✓ | | | ✓ |
| [78] | ✓ | | Business | COVID-19 challenges and I4.0 solutions | ✓ | ✓ | ✓ | ✓ |
| [79] | ✓ | | Business | Agile supply networks | | | ✓ | ✓ (Digital twins) |
| [80] | ✓ | | Business | Supplier selection and order allocation under disruptions | | | ✓ | ✓ |
| [44] | ✓ | | Food | Strategies to mitigate outbreak effects | | ✓ | | ✓ |
| [67] | ✓ | | Additive Manufacturing | Blockchain in Additive Manufacturing and its Impact on SCs | | | ✓ | ✓ (Blockchain) |
| [21] | ✓ | | | Global changing paradigms and sustainability issues | ✓ | | | ✓ |
| [81] | ✓ | | Business | Organizational resilience in changing times | ✓ | ✓ | | ✓ |
| [45] | ✓ | | Agriculture | Regional impact of SC disruptions | ✓ | ✓ | | ✓ |
| [53] | ✓ | | Logistics | Strategies for circular economy | ✓ | | ✓ | ✓ |
| [51] | | ✓ | Automotive | Improving SCR through I4.0 | ✓ | | | ✓ (Big Data) |
| [82] | ✓ | | Business | SCR using Big Data | | ✓ | ✓ | ✓ (Big Data) |
| [11] | ✓ | | Business | Coping with SC disruptions | | ✓ | | ✓ |
| [83] | ✓ | | Fashion | Opportunities and challenges with respect to I4.0 | ✓ | | | ✓ |
| [84,85] | | ✓ | Business | SC security | ✓ | | | ✓ (Blockchain) |
| [86] | ✓ | | Business | Inventory and cash management under physical and financial SC disruptions | | | ✓ | ✓ (Digital twins) |
| [22] | ✓ | | Manufacturing | Sustainable production alternatives | | ✓ | | ✓ |
| [67] | ✓ | | Additive Manufacturing | SCR and efficiency building | ✓ | ✓ | | ✓ |
| [57] | ✓ | | Automotive | SCR during COVID-19 | ✓ | | | ✓ |
| [87] | ✓ | | Manufacturing | Role of I4.0 and resilience in circular SCs | ✓ | ✓ | | ✓ |
| [23] | ✓ | | Business | Digital technologies and circular economy practices | | ✓ | | ✓ |

**Table 1.** *Cont.*

| References | Article Type | | Industry/Focus Area | Main Idea | Methodology | | | I4.0 Technologies Used/Recommended |
|---|---|---|---|---|---|---|---|---|
| | Research | Review | | | Conceptual/ Theoretical | Empirical | Mathematical/ Simulation | |
| [12] | ✓ | | Business | SCR during a crisis | ✓ | | | ✓ |
| [59] | ✓ | | Automotive | COVID-19 impact on SC operations | | ✓ | | ✓ |
| [18] | ✓ | | Business | Global changes and disruptions in SC | ✓ | | | ✓ |
| [39] | ✓ | | Maritime/Logistics | COVID-19 challenges | ✓ | | | ✓ |
| [34] | | ✓ | Maritime | SCR | ✓ | | | ✓ |
| [36] | ✓ | | Maritime/Port | Resilience analysis | ✓ | | | ✓ |
| [37] | ✓ | | Maritime/Port | Resilience enhancement through port coalitions | | | ✓ | |
| [50] | ✓ | | Logistics | Overcoming digitalization barriers for SCR | | | ✓ | ✓ |
| [19] | | ✓ | Business | SCR during COVID-19 | ✓ | | | ✓ |
| [88] | ✓ | | Manufacturing | I4.0 enablers to mitigate ripple effects | | ✓ | | ✓ |
| [83] | ✓ | | Textile | Resilient SC model | | ✓ | | ✓ |
| [25] | ✓ | | Business | Digital SC management and technology to enhance resilience | ✓ | ✓ | | ✓ |
| [24] | ✓ | | Business | Cloud SC | ✓ | | | ✓ |
| [32] | ✓ | | Healthcare | Procurement 4.0 | ✓ | | | ✓ |
| [46] | ✓ | | Agriculture | Digital technologies and food security | | | ✓ | ✓ |
| [89] | ✓ | | Business | Cybersecurity | ✓ | | | ✓ |
| [86] | ✓ | | Additive Manufacturing | SC disruptions from COVID-19 | ✓ | | | ✓ |
| [90] | ✓ | | Business | Digital SC management | ✓ | | ✓ | ✓ |
| [91] | ✓ | | Agriculture | Sustainability development in agricultural SC | ✓ | | | ✓ (Blockchain) |
| [92] | ✓ | | Business | SC viability in COVID-19 perspective | ✓ | | | ✓ |
| [26] | | ✓ | Business | Global SC in post-COVID-19 scenario | ✓ | | | ✓ |
| [36] | | ✓ | Food | Digitalization in food SCs | ✓ | | | ✓ |
| [93] | ✓ | | Automotive | LARG Manufacturing | | | ✓ | ✓ |
| [28] | ✓ | | Business | Survivability of sustainable SCs during and post-COVID-19 | | | ✓ | ✓ |
| [19] | ✓ | | Business | Sustainable development post COVID-19 disruptions | | ✓ | ✓ | ✓ |
| [47] | ✓ | | Agriculture | Agriculture and digital farming | ✓ | | | ✓ |
| [31] | ✓ | | Healthcare | Development of resilience and digitalization | | ✓ | | ✓ |
| [27] | ✓ | | Logistics | Digital logistics and SC management | ✓ | | ✓ | ✓ |
| [29] | ✓ | | Business | DT to achieve SCR | | ✓ | ✓ | ✓ |

## 6. Digital Supply Chain Resilience and Additive Manufacturing

In addition to the above-mentioned I4.0 technology enablers, additive manufacturing (AM) is one of the enablers that portray slightly different characteristics, as it combines automation and manufacturing. AM is technically defined as the process of building something up, and it typically refers to 3-D printing [92].

The role of AM in building SCR is regarded as instrumental in the wake of I4.0 and the post-COVID-19 scenario. Research emphasizes the importance of the improvement in SCR through promotional INV and distributed production by using AM and on SCR during a disruption in the diffusion of green products in a reverse logistics setup. In the same context, blockchain-based additive manufacturing can potentially make SCs more visible and digital, supporting its financing and the development of shared factory systems. However, the limitations of the process that need to be addressed include the nonexistence of blockchain-skilled labor in the market, the lack of technical expertise in companies, and the absence of proper policies [66,77–80].

In the African SC context, case studies and focus groups methodologies were employed with the objective of investigating the potential of AM in increasing both SCR and efficiency, which they call ambidextrous qualities. The research determined that the AM can reconcile these qualities by developing them at the SC level. The research predicts that investing in digital technologies such as AM to be substantial in simultaneously building SCR and efficiency in the post-COVID-19 era [86,89].

## 7. Securing the Digital Supply Chain: Cybersecurity and Emerging Technologies

The process of building DSCR is multifaceted, complex, expensive, and vulnerable; hence, a number of critical areas need to be carefully considered to ensure optimum results in, for example, cybersecurity, scalability, return on INV, among others [1]. In this

context, academia provides a comprehensive review about the importance of advancing the cybersecurity within the healthcare ecosystem, the research provides a focus on the use of self-optimizing and self-adaptive AI systems to enhance cybersecurity in the healthcare system. The research emphasizes the importance of continuous monitoring and adaptation of cybersecurity measures to address emerging threats and vulnerabilities. Similarly, a literature and bibliometric review conducts a comprehensive review of the literature on new and emerging forms of data and technologies, including big data analytics, blockchain, and the IoT. The research highlights the potential of these I4.0 technologies to improve SC resilience and cybersecurity, but also notes the need for careful consideration of ethical and legal issues. The authors identify several factors that influence the impact of AI on workplace outcomes, including the level of employee involvement in AI implementation, the degree of task complexity, and the level of trust and transparency in AI decision-making processes [94–97].

Overall, the study highlights the importance of a multi-process perspective when examining the impact of AI on workplace outcomes, and emphasizes the need for organizations to carefully consider the potential benefits and drawbacks of AI implementation in the workplace. While each of these research addresses different aspects of securing DSCR in cyber space, there are several common themes and insights that emerge, such as those in AlBanna et al.'s work [1]. One of the key themes is the importance of leveraging emerging technologies such as big data analytics, blockchain, and AI to enhance cybersecurity and improve SCR. Another important theme is the need for continuous monitoring and adaptation of cybersecurity measures to address emerging threats and vulnerabilities. Furthermore, all these articles emphasize the importance of collaboration and information sharing among stakeholders in the SC ecosystem, including government agencies, industry partners, and cybersecurity experts. This highlights the need for a holistic approach to securing DSCR, one that involves a wide range of actors and leverages the latest technologies and best practices.

In summary, enhancing cybersecurity and improving SCR in the cyber space by leveraging emerging technologies and adopting a collaborative and adaptive approach is a critical aspect to maintaining a secured DSCR that enables organizations to detect, avoid, manage and recover from disruptions.

## 8. Conclusions

The world of business is in a constant state of evolution, driven by fast-paced technological advancements and the creation of vast amounts of data, as well as unprecedented disruptions and vulnerabilities. In this context, the adoption of Industry 4.0 technologies and adopting DSCR has become a critical necessity to enable organizations to detect, avoid, manage, and recover from disruptions efficiently.

This research has presented a bibliometric investigation and analysis that reveals that investment in Industry 4.0 is a complex, multi-faceted and expensive project. The study highlights that a successful DSCR implementation is not solely reliant on appropriate technologies but also on the right strategies for their adoption and the optimum amounts of investment in selected technologies. Over-investment as well as under-investment can lead to undesirable outcomes, as one leads to eroding the organization's profits, while the other exposes the organization to increased risks and vulnerabilities.

Furthermore, the analysis shows that digital transformation serves as a bridge to take organizations from their current state to their desired future state. However, organizations need to adopt a comprehensive approach that considers current vulnerabilities, future requirements, and environmental attributes, such as regulatory affairs, scalability, and cybersecurity, among others.

The convergence of Industry 4.0 and DSCR has emerged as a vital topic in recent years, with the potential to transform organizations across multiple industries. This research presents a bibliometric analysis of the interplay between Industry 4.0 and DSCR under INV

constraints, revealing trends, gaps, and opportunities for future research on the SCR-I4.0–INV interconnectedness.

To successfully implement DSCR, organizations must embrace a comprehensive approach that considers the right technologies, strategies, and environmental attributes. By carrying this out, companies can remain agile, competitive, and prepared to face unexpected disruptions and vulnerabilities in a rapidly changing business landscape. Additionally, organizations must continue to explore and search for innovative Industry 4.0 technology solutions, as today's innovations will soon become obsolete and outdated solutions.

**Author Contributions:** Conceptualization, A.A.-B. and B.M.; methodology, A.A.-B. and B.M.; data analysis visualization, writing—original draft preparation, A.A.-B. and Z.A.R.; validation, writing—review and editing, A.A.-B., M.Y. and B.M.; supervision, overall guidance, project management, B.M. All authors have read and agreed to the published version of the manuscript.

**Funding:** This research received no external funding.

**Data Availability Statement:** Data are unavailable due to privacy restrictions.

**Conflicts of Interest:** The authors declare no conflict of interest.

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
