# Peer review of "Interconnectedness between Supply Chain Resilience, Industry 4.0, and Investment"

_logistics, 2023_

Round 1

Reviewer 1 Report

Dear The Authors.

I have done review manuscript ID Logistic-2377883 "Interconnectedness between Supply Chain Resilience, Industry 2 4.0, and Investment"

There is strong paper argument ( relationship to literature, methodology, Implications for the research, practice and/or society and Quality communication).

Therefore I kindly recommend that this paper to be accept.

Thank you very much for the opportunity to invite me as a peer reviewer for the journal of MDPI.

Sincerely,

Yoyok Cahyono

Reviewer

Dear The Authors.

I have done review manuscript ID Logistic-2377883 "Interconnectedness between Supply Chain Resilience, Industry 2 4.0, and Investment"

Quality of English is very easy to understand.

Therefore I kindly recommend that this paper to be minor editing of English language required.

Sincerely,

Yoyok Cahyono

Reviewer

Reviewer 2 Report

The submitted paper presents Bibliometric Analysis of Interconnectedness between Supply Chain Resilience, Industry 2 4.0, and Investment  Although the paper might have some novelties, some points need clarification:

1) What are the Limitations of the current literature review ?

2) What could be the propodsed  solution ?

3) Scopus and Wos is considered for the study. Comparative analysis can be shown in the graphs.

4) There are some typos that should be corrected. It is recommended the authors review the paper again. Some examples are as follows: dot before conclusion, 

5) there are some Grammatically corrections

6) There are issues with flow of the paper 

there are some Grammatically corrections

Reviewer 3 Report

This submitted manuscript contributes to the growing body of literature on the intersection of Industry 4.0 and supply chain.

The submitted manuscript provides insights into how organizations can use digital technologies to enhance their resilience and prepare for unexpected disruptions.

The submitted manuscript presents a bibliometric investigation and analysis of the implementation of Industry 4.0 and supply chain. Some of the analysis presented in the submitted manuscript reveals that investment in Industry 4.0 is a complex, multi-faceted and expensive project. 

The study highlights that a successful implementation is not solely reliant on appropriate technologies but also on the right strategies for their adoption and the optimum amounts of investment in selected technologies. 

The paper is well-written and the authors did a good job of compiling the appropriate papers with the right analysis. 

The authors, nonetheless, did not include enough recent papers published in the year 2022 and 2023, in fact, only 3 cited papers are published in 2023. I recommend the authors update the literature review by adding some relevant papers on the topic of supply chains such as:

"Smart supply chain management in Industry 4.0: the review, research agenda and strategies in North America." Annals of Operations Research 322.2 (2023): 1075-1117.

"Routing and resource allocation in non-profit settings with equity and efficiency measures under demand uncertainty" Transportation Research Part C: Emerging Technologies, 149 (2023): 104023. 

"Industry 4.0 technology capabilities, resilience and incremental innovation in Australian manufacturing firms: a serial mediation model." Supply Chain Management: An International Journal ahead-of-print (2023).

"A multi-objective home healthcare delivery model and its solution using a branch-and-price algorithm and a two-stage meta-heuristic algorithm" Transportation Research Part C: Emerging Technologies, (2022): 103838. 

 "Cyber supply chain risk management and performance in industry 4.0 era: information system security practices in Malaysia." Journal of Industrial and Production Engineering 40.2 (2023): 102-116.

"A systems approach to carbon policy for fruit supply chains: Carbon tax, technology innovation, or land sparing?." Science of The Total Environment 767 (2021): 144211.

Round 2

Reviewer 2 Report

All reviewers' comments have been addressed. The manuscript can now be accepted.